# Deep Pre-Alignment for VLMs

**Tianyu Yu**[1]  **Kechen Fang**[1]  **Zihao Wan**[1]  **Kaidong Zhang**[1]  **Yicheng Zhang**[1]  **Jun Song**[2]  **Bo Zheng**[2]
**Yuan Yao**[1][3]

yiranytianyu@gmail.com    yaoyuanthu@gmail.com

DPA Code and Model

## Abstract

Most Vision Language Models (VLMs) directly map outputs from ViT encoders to the LLM via a lightweight projector. While effective, recent analysis suggests this architecture suffers from an alignment challenge: visual features remain distant from the text space in the initial layers of the LLM, forcing the model to waste critical depth (Zhang et al., 2024; Artzy & Schwartz, 2024) on superficial modality alignment rather than deep understanding and complex reasoning. In this work, we propose Deep Pre-Alignment (DPA), a novel architecture that replaces the standard ViT encoder with a small VLM as perceiver, ensuring visual features are deeply aligned with the text space of the target large language model. Comprehensive experiments demonstrate the effectiveness of DPA. On the 4B parameter scale, DPA outperforms baselines by 1.9 points across 8 multimodal benchmarks, with gains widening to 3.0 points at the 32B scale. Moreover, by offloading alignment to the perceiver, DPA achieves a 32.9% reduction in language capability forgetting over 3 text benchmarks. We further demonstrate that these gains are consistent across different LLM families including Qwen3 and LLaMA 3.2, highlighting the generality of our approach. Beyond performance, DPA also offers a seamless upgrade path for current VLM development, requiring only a modular replacement for the visual encoder with marginal computation overhead.

---

[1]Tsinghua University [2]Taobao & Tmall Group of Alibaba [3]Shanghai Qi Zhi Institute. Correspondence to: Yuan Yao <yaoyuanthu@gmail.com>.

*Proceedings of the 43rd International Conference on Machine Learning*, Seoul, South Korea. PMLR 306, 2026. Copyright 2026 by the author(s).

## 1. Introduction

The surge of Vision Language Models (VLMs) has been driven by a simple and effective architecture that connects a pre-trained ViT encoder to a Large Language Model (LLM) via a lightweight projector (Bai et al., 2023; Wang et al., 2024b; Yao et al., 2024; Yu et al., 2025; Bai et al., 2025a; Cui et al., 2026). The underlying assumption is that these ViT encoders are sufficiently aligned with language to serve as direct inputs for the LLM, requiring only a linear or MLP projection to bridge the dimensions.

However, recent studies on the internal representation dynamics of VLMs suggest that pre-trained visual encoder features still suffer from a significant modality gap to large language models' text space (Huang et al., 2025). Consequently, the burden of modality alignment is transferred to the large language model layers, which progressively align distant multimodal features with text space. Nikankin et al. (2025) also find that VLMs use different neuron circuits for visual and textual inputs in low-level layers, drawing the concern that current VLMs may waste critical model depth on superficial modality alignment objective. These phenomena imply that these initial layers are diverted from their primary role, creating a distraction that triggers the catastrophic forgetting of language capabilities that commonly observed in VLMs (McKinzie et al., 2024; Lu et al., 2024a; Bai et al., 2025b).

To resolve this inefficient allocation of depth, we propose the Deep Pre-Alignment (DPA) architecture, which replaces the standard ViT encoder with a small VLM as perceiver. By leveraging the perceiver's internal layers to progressively bridge the modality gap upstream, DPA ensures that visual features are deeply aligned with the text space before entering the target model. Consequently, the language model requires less destructive adaptation and is able to better inherit the strong skills obtained from language pre-training to help text and multimodal understanding. Critically, DPA provides a seamless upgrade path for modern VLM development, since it only requires a modular replacement for the visual encoder with the training objective and inference strategy untouched. This advantage makes it easy to implement in commonly used frameworks.

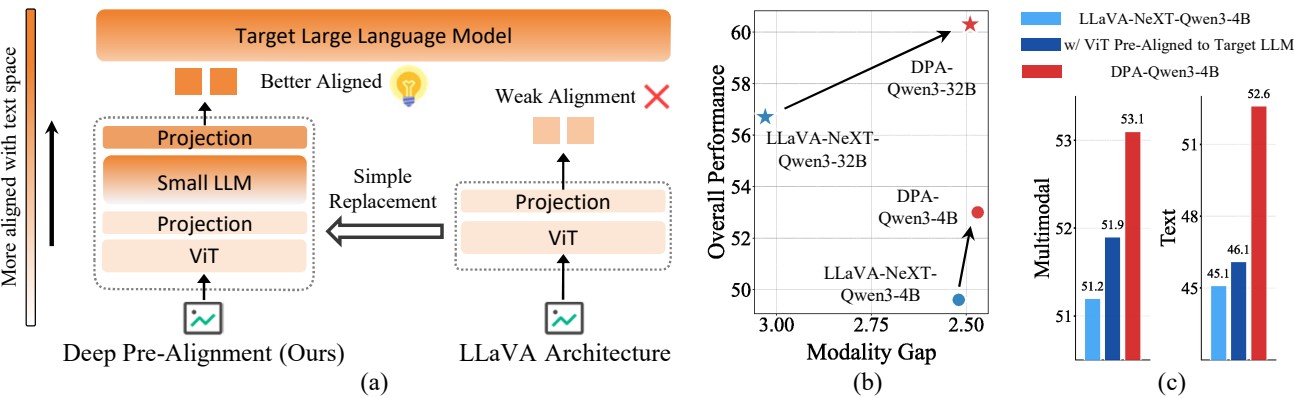

*Figure 1.* (a) **Architectural overview of DPA.** By simply replacing the ViT encoder with a perceiver VLM, DPA offloads the superficial modality alignment burden from the target large language model, and deeply align visual features inside the perceiver language blocks. The input visual features are thus better aligned with text space. (b) DPA significantly minimize the modality gap (Huang et al., 2025) and also improves the overall performance over 11 popular benchmarks. (c) While pre-align the ViT encoder with the target LLM improves the performance, DPA achieves a significantly higher improvement, suggesting the effectiveness of the architectural design.

We extensively validate DPA across 8 popular multimodal benchmarks and 3 popular text benchmarks, covering general visual understanding, fine-grained perception, multimodal reasoning and text-only tasks. Comprehensive experimental results demonstrate that DPA consistently outperforms the standard LLaVA (Liu et al., 2024a) architecture across all domains. Notably, our method exhibits positive scalability: the average multimodal performance gain widens from 1.9 points at the 4B scale to 3.0 points at the 32B scale. We confirm the universality of DPA by demonstrating that these gains persist across different LLM families, specifically Qwen3 (Yang et al., 2025) and LLaMA 3.2 (Meta, 2024). Furthermore, by offloading superficial alignment duties from the target large language model, DPA significantly mitigates catastrophic forgetting, reducing text performance degradation by 32.9% and 21.6% in 4B and 32B models. Importantly, these gains are achieved with negligible computation overhead relative to the target large language model size (e.g., only 2% throughput loss on 32B models). Finally, analysis of internal representations confirms that DPA not only significantly bridge the modality distance (Huang et al., 2025) but constructs a visual feature space that is geometrically compatible with the target text space. Codes, data and model weights will be released to facilitate further research and application.

## 2. Deep Pre-Alignment

We now introduce the details of Deep Pre-Alignment. The key of our design is to decouple the alignment objective from the deep understanding and complex reasoning objectives of large language models by offloading the alignment burden to a dedicated upstream module.

### 2.1. Architectural Overview

The DPA architecture consists of three distinct components: a perceiver VLM module ($M_p$), an alignment projector ($\phi$) and a target large language model ($M_t$). The perceiver $M_p$ itself contains a ViT module ($\mathcal{E}$), projector ($\phi_p$) and language model blocks ($M_p^{\text{LLM}}$).

Standard MLLM architectures (Liu et al., 2024a) typically follow a serial pipeline:

$$v \xrightarrow{\mathcal{E}} \mathbf{H}_v \xrightarrow{\phi} \mathbf{H}'_v \to M_t \qquad (1)$$

where a ViT encoder $\mathcal{E}$ (e.g., CLIP ViT (Radford et al., 2021) ) outputs visual features $\mathbf{H}_v$. Since $\mathbf{H}'_v$ is still distant from the text space of the target large language model, this architecture places strong modality alignment burden on initial layers.

In contrast, DPA replaces the ViT encoder $\mathcal{E}$ with a small perceiver VLM $M_p$:

$$v \xrightarrow{\mathcal{E}} \mathbf{H}_v \xrightarrow{\phi_p} \mathbf{H}'_v \xrightarrow{M_p^{\text{LLM}}, \phi} \mathbf{H}_{\text{aligned}} \to M_t \qquad (2)$$

As shown in Figure 1, $M_p$ progressively refines visual features generated by the ViT encoder through deep language blocks to pre-align the features with text space. By utilizing better text-aligned visual features, the target large language model requires less destructive adaptation and is able to better inherit the strong skills obtained from language pre-training to help text and multimodal understanding.

### 2.2. Alignment via Perceiver Language Blocks

The core innovation of DPA lies in leveraging the perceiver's internal language blocks to bridge the modality gap upstream of the target LLM. Unlike standard ViT encoders

from CLIP models, which achieve only shallow alignment with text space, the perceiver incorporates language blocks pre-trained on large-scale causal language modeling tasks. By extracting the hidden states of visual tokens from the final layer of $M_p^{\text{LLM}}$, we obtain visual features that are structurally adapted to the generative process of language models. This pre-alignment significantly reduces the need for destructive adaptation within the target LLM.

### 2.3. Training Paradigm

A distinct advantage of DPA is its seamless compatibility with established VLM frameworks. Thanks to its simple architectural design, we eliminate the need for complex auxiliary loss of specialized training strategy. Instead, DPA is trained using the standard language modeling objective and the widely adopted two-stage training recipe (Liu et al., 2024a;b). In stage-1, we updates only the projection layer $\phi$ with image-caption data to bridge the dimension gap between the perceiver $M_p$ and the target large language model $M_t$. In stage-2, we fine-tune the entire model end-to-end on high-quality visual instruction data.

## 3. Experiments

In this section, we empirically investigate the effectiveness of DPA across various model scales and model families. We first introduce the experimental setup and then analyze the main experimental results in detail.

### 3.1. Experimental Setup

**Models.** We mainly conduct experiments using Qwen3 (Yang et al., 2025) as the target large language model, validating the effectiveness of DPA on both the 4B and 32B parameter scales. We additionally run experiments on LLaMA 3.2 (Meta, 2024) to test whether DPA is effective regardless of language model families. We compare our trained DPA models with two distinct categories of baselines:

- **Controlled Baseline:** To rigorously isolate the impact of DPA, we construct controlled baselines named LLaVA-NeXT-Qwen3-4B, LLaVA-NeXT-Qwen3-32B and LLaVA-NeXT-LLaMA-3.2-3B following LLaVA-NeXT (Liu et al., 2024b) architecture with ViT from (Yang et al., 2025). These models are identical to their DPA counterparts in every aspect (training data, scheduler, target LLM) except not using a perceiver for deep pre-alignment.

- **Public Baselines:** To benchmark against the broader landscape, we also report performance for established open-source VLMs including LLaVA-1.5-7B (Liu et al., 2024a), LLaVA-NeXT-7B (Liu et al., 2024b), Qwen2-VL-2B (Wang et al., 2024b), Idefics2-

8B (Laurençon et al., 2024) and Cambrian-1-8B (Tong et al., 2024). We also compare the textual performance with Qwen3 (Yang et al., 2025) language models to access the text capability forgetting level. Note that these models vary in training data and initialization.

**DPA Configuration.** Our method replaces the standard ViT encoder of controlled baselines with a compact Qwen3-0.6B perceiver VLM that utilizes the same underlying ViT. We train the perceiver using the exact same hyperparameter settings as the controlled baselines. Furthermore, we validate DPA using untrained perceivers, demonstrating that the architecture yields strong performance improvements even without pre-existing of trained perceiver, as detailed in Section 4.2.

**Training Data.** We train both the controlled baselines and DPA models on the exact same two-stage corpus: 558K web-source image-caption pairs from (Liu et al., 2024a) for alignment, followed by 1M high-quality visual instruction samples from (Guo et al., 2025) for instruction tuning.

**Benchmarks.** We evaluate models on a comprehensive suite of 11 benchmarks covering four domains: (1) **General.** We access the general visual understanding performance with SeedBench2Plus (Li et al., 2024), MM-Vet (Yu et al., 2023), and MMStar (Chen et al., 2024a), covering both free-form and multiple-choice-question tasks. (2) **Reasoning.** We measure the multimodal reasoning capabilities on MMMU (Yue et al., 2024) for multi-discipline reasoning, and on MathVista (Lu et al., 2024b) and MathVision (Wang et al., 2024a) for math reasoning. (3) **Perception.** We evaluate the fine-grained perception capability on OCRBench (Liu et al., 2024c) for text recognition and AI2D (Kembhavi et al., 2016) for diagram interpretation. (4) **Text.** We evaluate the text task performance on MATH-500 (Cobbe et al., 2021), MMLU-Redux (Gema et al., 2025) and GPQA-diamond (Rein et al., 2024), covering mathematical reasoning and general multi-discipline knowledge.

**Implementation Details.** In stage-1, we adopt a maximum learning rate of 1e-3, each batch contains 512 samples. In stage-2, we adopt a maximum learning rate of 1e-5, and each batch contains 256 samples. For both stages, we train the model for 2 epochs with cosine learning rate decay strategy. For the 32B setting, we employ LoRA (Hu et al., 2022) to maintain computation feasibility and train for 3 epochs in instruction tuning stage for better convergence. More implementation details are listed in the Appendix.

### 3.2. Main Results

The main experimental results are reported in Table 1. We observe that DPA delivers a consistent and robust performance improvement on all tasks groups, model scales and model families.

*Table 1.* **Main experimental results.** We report performance grouped by capability (General/Reasoning/Perception/Text), where **Avg.** denotes the average score within each capability group; ***Multi. Avg.*** denotes the average over the 8 multimodal benchmarks; and ***All Avg.*** denotes the average over all 11 benchmarks. We include △ rows showing the improvement of our **DPA** models over controlled baseline, with better results highlighted in **bold**.

| Method | General | | | | Reasoning | | | | Perception | | | Text | | | | *Multi.* *Avg.* | *All* *Avg.* |
|---|---|---|---|---|---|---|---|---|---|---|---|---|---|---|---|---|---|
| | MMVet | MMStar | Seed Bench2+ | Avg. | MMMU | Math Vista | Math Vision | Avg. | OCR Bench | AI2D | Avg. | MATH 500 | MMLU Redux | GPQA Diamond | Avg. | | |
| **Public Models** | | | | | | | | | | | | | | | | | |
| LLaVA-1.5-7B | 31.1 | 32.3 | 41.2 | 34.9 | 34.9 | 24.1 | 12.2 | 23.7 | 20.2 | 52.8 | 36.5 | 2.6 | 42.6 | 12.6 | 19.3 | 31.1 | 27.9 |
| LLaVA-NeXT-7B | 39.0 | 37.7 | 50.5 | 42.4 | 35.1 | 34.4 | 12.6 | 27.4 | 50.2 | 66.6 | 58.4 | 1.6 | 43.6 | 12.1 | 19.1 | 40.8 | 34.9 |
| Qwen2-VL-2B | 49.5 | 48.0 | 62.3 | 53.3 | 41.1 | 43.0 | 12.4 | 32.2 | 80.9 | 74.7 | 77.8 | 19.2 | 45.3 | 15.7 | 26.7 | 51.5 | 44.7 |
| Idefics2-8B | 31.5 | 49.4 | 56.9 | 45.9 | 43.1 | 52.2 | 13.6 | 36.3 | 63.0 | 72.5 | 67.8 | 13.8 | 50.0 | 15.2 | 26.3 | 47.8 | 41.9 |
| Cambrian-1-8B | 44.2 | 50.9 | 60.4 | 51.8 | 40.8 | 47.0 | 22.2 | 36.7 | 60.2 | 74.6 | 67.4 | 0.2 | 8.1 | 0.0 | 2.8 | 50.0 | 37.1 |
| Qwen3-4B | - | - | - | - | - | - | - | - | - | - | - | 84.8 | 77.3 | 41.7 | 67.9 | - | - |
| Qwen3-32B | - | - | - | - | - | - | - | - | - | - | - | 88.6 | 85.7 | 54.6 | 76.3 | - | - |
| **LLaMA-3.2-3B as Target LLM** | | | | | | | | | | | | | | | | | |
| LLaVA-NeXT-LLaMA-3.2-3B | 30.8 | **41.0** | 50.5 | 40.8 | 32.9 | 34.0 | **15.4** | 27.4 | 62.0 | 59.0 | 60.5 | 4.8 | 41.0 | 17.2 | 21.0 | 40.7 | 35.3 |
| **DPA-LLaMA-3.2-3B** | **42.0** | 40.7 | **51.7** | **44.8** | **37.8** | **36.7** | 15.0 | **29.8** | **69.2** | **59.5** | **64.3** | **6.4** | **43.6** | **25.3** | **25.1** | **44.1** | **38.9** |
| △ | +11.1 | -0.3 | +1.1 | +4.0 | +4.9 | +2.7 | -0.4 | +2.4 | +7.2 | +0.5 | +3.8 | +1.6 | +2.6 | +8.1 | +4.1 | +3.4 | +3.6 |
| **Qwen3-4B as Target LLM** | | | | | | | | | | | | | | | | | |
| LLaVA-NeXT-Qwen3-4B | 41.4 | 50.7 | 61.2 | 51.1 | 49.9 | 49.6 | 20.7 | 40.1 | 69.1 | 67.4 | 68.3 | 36.4 | 66.0 | 32.8 | 45.1 | 51.2 | 49.6 |
| **DPA-Qwen3-4B** | **42.7** | 50.7 | **64.0** | **52.5** | **50.0** | **51.8** | **21.2** | **41.0** | **74.7** | **70.0** | **72.4** | **54.2** | **70.2** | **33.3** | **52.6** | **53.1** | **53.0** |
| △ | +1.3 | -0.0 | +2.8 | +1.4 | +0.1 | +2.2 | +0.5 | +0.9 | +5.6 | +2.6 | +4.1 | +17.8 | +4.2 | +0.5 | +7.5 | +1.9 | +3.4 |
| **Qwen3-32B as Target LLM** | | | | | | | | | | | | | | | | | |
| LLaVA-NeXT-Qwen3-32B | 49.2 | 56.1 | 67.5 | 57.6 | 56.9 | 61.8 | 26.3 | 48.3 | 73.8 | 73.0 | 73.4 | 44.4 | 79.5 | 35.3 | 53.1 | 58.1 | 56.7 |
| **DPA-Qwen3-32B** | **56.0** | **59.1** | **67.7** | **60.9** | **57.1** | **63.7** | **29.4** | **50.1** | **76.8** | **79.0** | **77.9** | **58.2** | **80.6** | **35.4** | **58.1** | **61.1** | **60.3** |
| △ | +6.8 | +3.0 | +0.2 | +3.3 | +0.2 | +1.9 | +3.1 | +1.8 | +3.0 | +6.0 | +4.5 | +13.8 | +1.1 | +0.1 | +5.0 | +3.0 | +3.6 |

**Multimodal Capabilities.** As shown in Table 1, DPA achieves superior performance on all multimodal tasks groups compared with controlled baselines, validating that the deep alignment process inside the perceiver preserves critical visual information for multimodal tasks. For example, in the perception domain, DPA improves the performance on AI2D for 2.6 and 6.0 points on the 4B and the 32B scales separately.

**Mitigating Text Capability Forgetting.** As shown in the Text column of Table 1, VLMs exhibit severe degradation in text-only task performance. For example, the MATH-500 performance of the 4B baselines drops to 36.4 from 84.8. While DPA does not fully eliminate this phenomenon, it significantly mitigates the forgetting by 17.8 points (36.4 → 54.2). The advantage patterns holds across multiple text benchmarks, resulting in an overall forgetting reduction of +7.5 points (32.9% relatively) at the 4B scale and +5.0 points (21.6% relatively) at the 32B scale. These improvements suggest that DPA effectively minimizes destructive parameter adaptation.

**Positive Scalability.** Comparing the multimodal performance improvement of different model scales, we find that the lift expands on all domains from 4B to 32B on general visual understanding (+1.4 →+3.3), multimodal reasoning (+0.9 →+1.8), and fine-grained perception (+4.1 →+4.5).

These results suggest that the lack of deep pre-alignment is a persistent bottleneck in standard architecture that cannot be solved by merely scaling the target LLM sizes.

**Universality across Model Families.** To verify that the effectiveness of DPA is not tied to a specific large language model family, we additionally validate DPA with LLaMA-3.2-3B (Meta, 2024) as the target large language model. Results in Table 1 demonstrate that, under the same training setting, DPA-LLaMA-3.2-3B consistently outperforms the corresponding controlled baseline across general understanding, reasoning, perception, and text-only tasks, yielding a +3.6 overall average gain. This confirms that the benefit of deep pre-alignment is model-agnostic and transfers across different LLM families.

## 4. Analysis

Having demonstrated the performance and scalability of DPA, we now analyze the design choices and internal mechanisms driving these gains. We structure our analysis as five key research questions:

- **(RQ1) Perceiver Performance:** How does the standalone performance of the perceiver impact the final performance of DPA models?

*Table 2.* **Perceiver standalone evaluation.** We train multiple perceiver VLMs using different number of training samples with the same recipe. ✓ / ×: the stage is applied or not. Best results highlighted in **bold**.

| Perceiver Stage-1 | Stage-2 | Gen. | Reas. | Perc. | Text | Avg. |
|---|---|---|---|---|---|---|
| × | × | 0.0 | 0.0 | 0.0 | **40.2** | 10.8 |
| ✓ | × | 9.1 | 14.2 | 19.8 | **40.2** | 21.1 |
| ✓ | 10k | 24.2 | 22.9 | 30.6 | 30.8 | 26.8 |
| ✓ | 250k | 30.7 | 25.5 | 45.3 | 24.2 | 30.2 |
| ✓ | 1M | **35.1** | **27.3** | **53.6** | 22.8 | **33.0** |

*Table 3.* **Performance of models using different perceivers.** LLaVA-Qwen: LLaVA-NeXT-Qwen3-4B model; DPA-Qwen: DPA-Qwen3-4B model; ✓ / ×: the stage is applied or not. Best results highlighted in **bold**.

| Model | Perceiver Stage-1 | Stage-2 | Gen. | Reas. | Perc. | Text | Avg. |
|---|---|---|---|---|---|---|---|
| LLaVA-Qwen. | – | – | 51.1 | 40.1 | 68.3 | 45.1 | 49.6 |
| DPA-Qwen. | × | × | 53.1 | 40.2 | 69.5 | 55.1 | 53.1 |
| | ✓ | × | **53.7** | 41.4 | 70.2 | 53.4 | 53.3 |
| | ✓ | 10K | 52.8 | 41.0 | 70.9 | **56.3** | 53.8 |
| | ✓ | 250K | 53.5 | **41.7** | 71.0 | 55.8 | **54.1** |
| | ✓ | 1M | 52.5 | 41.0 | **72.4** | 52.6 | 53.0 |

- **(RQ2) Essential Properties:** What are essential properties for the perceiver to bridge the modality gap?

- **(RQ3) Design Strategy:** How different design strategy on perceiver affect the efficacy of DPA?

- **(RQ4) Representation Dynamics:** How does DPA fundamentally alter the visual representations compared to standard architecture models?

- **(RQ5) Computational Efficiency:** What is the computational cost of DPA compared to standard architecture using ViT as encoder?

## 4.1. Impact of Perceiver Performance

It is intuitive to assume that a stronger perceiver yields higher DPA model performance. To test this hypothesis, we train a series of perceiver models with varying degrees of costs: ranging from an untrained baseline (where the $\phi_p$ and $\phi$ are randomly initialized) to a full version trained on 1M instruction samples. Perceivers of all setting except the untrained baselines undergoes a full stage-1 training using the 558K image-caption pairs. The standalone performance of these perceiver VLMs spans a wide spectrum from as detailed in Table 2, where the multimodal performance continuously increases with more training. We observe that the text performance continuously drops (from 40.2 to 22.8), strengthening the text capability forgetting problem of VLMs. We then leverage each perceiver to train a final DPA based 4B VLM under identical settings as in the main

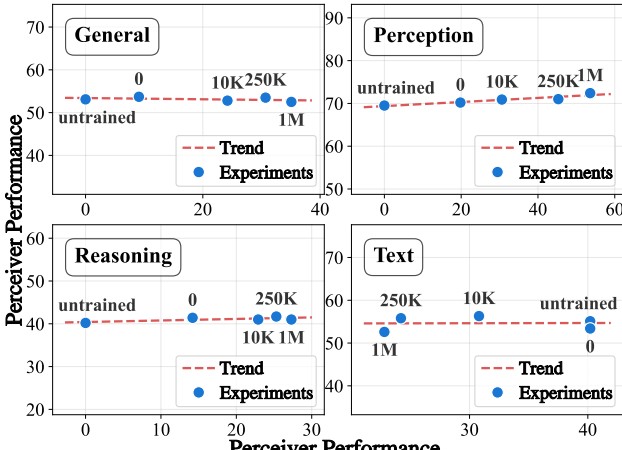

*Figure 2.* **Correlation between perceiver standalone performance and corresponding DPA model performance**. $\rho$ denotes the Pearson correlation coefficient in each task group. 0, 10K, 250K, 1M: number of instruction samples used to train corresponding perceivers; untrained: the perceiver used in DPA consists of randomly initialized projections.

experiments. The evaluation results of these models are shown in Table 3.

**Robust Improvement Regardless of Perceiver Performance.** Results in Table 3 show that even untrained perceiver baseline, which has randomly initialized projection layers, achieves +3.5 points overall performance improvement over the standard ViT encoder based architecture. Though perceiver performance ranges from 10.8 to 33.0 points (multimodal average from 0 to 36.8 points), the overall average performance improvement remains stable in around 3.4 to 4.5 points, suggesting that key effectiveness roots from the architectural deep pre-alignment design rather than transfer capability from perceiver models. We further investigate the perceiver performance effect on each capability domain and compute the Pearson correlation coefficients. Results shown in Figure 2 reveals that the correlation on general visual understanding and text performance is neglectable. Moreover, though positive correlation is observed on perception and reasoning domains, the final effect magnitude is very limited. For example, when perceiver reasoning score increases from 0.0 to 27.3, the outcome reasoning score improvement is only 0.8 points.

## 4.2. Essential Properties of the Perceiver

Having established that the performance of perceiver is not a decisive factor, we now investigate what specific properties make the perceiver effective. We conduct ablation studies using the weights from our full perceiver trained with 1M visual instructions.

**Language Blocks are Necessary.** We compare the default DPA configuration against a "*w/o* perceiver LM blocks" variant, where we remove the language blocks from the per-

*Table 4.* **Ablation on perceiver components and design strategies.** Best results highlighted in **bold**.

| Model | Gen. | Reas. | Perc. | Text | Avg. |
|---|---|---|---|---|---|
| LLaVA-NeXT-Qwen3-4B | 51.1 | 40.1 | 68.3 | 45.1 | 49.6 |
| *w/* large MLP | 26.2 | 29.7 | 29.2 | 49.8 | 34.1 |
| DPA-Qwen3-4B | 52.5 | 41.0 | **72.4** | 52.6 | 53.0 |
| *w/o* perceiver LM blocks | 51.7 | 40.3 | 69.5 | 46.1 | 50.3 |
| *w/o* perceiver LM pre-training | 30.4 | 32.6 | 32.2 | 57.5 | 38.7 |
| *w/* instruction context | 53.8 | **41.8** | 71.8 | **59.0** | **55.2** |
| *w/* perceiver frozen | 51.7 | 39.9 | 67.4 | 54.4 | 52.1 |
| *w/* untrained perceiver | 53.1 | 40.2 | 69.5 | 55.1 | 53.1 |
| *w/* untrained perceiver + multi-task | **55.3** | 41.3 | 71.0 | 53.6 | 53.9 |

ceiver and use only the ViT weights trained in the perceiver. The results in Table 4 show that even the ViT weight is pre-aligned to Qwen3-0.6B LLM, the performance improvement is still marginal and significantly less than using the full perceiver as visual encoder (+0.7 points vs. +3.4 points). This confirms that deep language blocks of perceiver are necessary and contribute most performance improvement.

**LLM Initialization Matters.** Based on the effectiveness of the untrained setting results in Table 3, we derive a fundamental conclusion: The perceiver does not need to possess pre-existing multimodal capability. A intuitive question is that whether language capability is required. To answer this, we further replace the pre-trained weight inside the language blocks $M_p^{\text{LLM}}$ with randomly weight, and conduct training for the same setting as main experiments. Results in Table 4 shows that language pre-training is essential property for perceiver as the "*w/* perceiver LM pre-training" performance drops significantly.

### 4.3. Design Strategy

We analyze the effect of different perceiver design strategies regarding its inputs and training specification.

**Early Fusion Further Mitigates Text Capability Forgetting.** A unique capability of fully functional VLM as encoder compared with ViT is that it can understand images and instructions together. This allows the perceiver to perform early fusion and dynamically extract visual features specifically relevant to the input query. We ablate this design choice by comparing our default instruction-agnostic schema as shown in Figure 1 against a "*w/* instruction context" variant in Table 4. As shown in the table, conditioning visual feature encoding on the textual instruction during both training and evaluation raise the overall performance from 53.0 to 55.2. Specifically, most improvement comes from text capability, where the score increases from 52.6 to 59.0. We hypothesize that the textual instruction passed to the perceiver naturally acts as a early fusion semantic filter, suppressing visual features that are irrelevant to the current

query. This further mitigate the destructive adaptation of the target large language model and consequently help retaining more language capabilities.

**Precision vs. Generalization Trade-off.** Despite the superior metrics of the context-aware visual encoding schema based on instructions, we keep removing instruction from perceiver inputs as our default choice to ensure generalization in real-world scenarios. While context-aware visual encoding significantly enhances single-turn performance on benchmarks, it inherently binds the visual features to a specific question, which can degrade proficiency in multi-turn dialogues where the user's intent shifts. Our results confirm that the default visual encoding schema of DPA models achieves superior performance compared with controlled baselines even without this dependency, offering a more robust solution for general-purpose VLMs, while the context-aware variant remains a powerful option for specialized and single-turn tasks.

**Perceiver Adaptation During Training Helps.** Furthermore, we conduct an ablation to study whether perceiver requires adaptation during the instruction tuning stage or not. Results in Table 4 show that DPA with frozen perceiver still surpasses the controlled baseline by +2.5 points. However, the performance is still obviously decreased compared with the default setting which keeps the perceiver trainable during instruction tuning stage, suggesting end-to-end fully tuning is more robust default strategy.

**Multi-Task Supervision Boosts Untrained Perceiver.** Although the untrained perceiver shows architectural promise, it still lags in fine-grained perception as shown in Table 3. To bridge this gap, we apply an auxiliary language modeling loss directly to the perceiver, treating it as a standalone VLM. This explicit supervision enforces semantic visual understanding within the perceiver, complementing the back-propagation from the target LLM. Results in Table 4 show that such multi-task training strategy (i.e., *w/* untrained perceiver + multi-task) substantially improves all multimodal performance. Empirically, we observe that applying this auxiliary loss is beneficial during stage-1 but slightly hinders performance if continued into stage-2. So we report results using multi-task supervision in stage-2 only.

### 4.4. Representation Dynamics

We investigate how DPA alters the visual representation space in this section. We use the MIR metric (Huang et al., 2025), which considers both distribution center distance and shape difference, to measure the modality gap.

**Modality Gap Bridges within the Perceiver.** First, we analyze the internal dynamics of standalone perceiver VLMs. Figure 3 illustrates the modality gap between each layer and the text space of a Qwen3 0.6B model. Note that here we

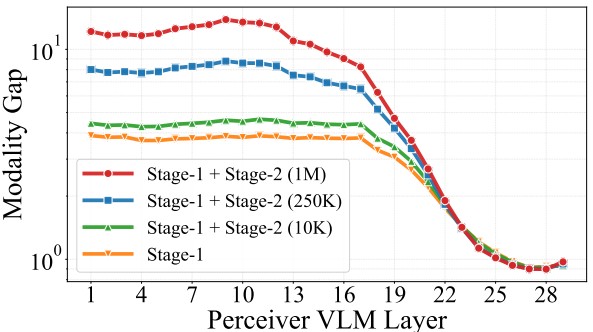

*Figure 3.* **Modality gap comparison of different perceiver layers.** We compute the layer-wise MIR between per-layer output of perceiver with the text space of the Qwen3 0.6B model. All models exhibit fast convergence of the modality gap in deep layers, and finally reach a similar level.

*Table 5.* **Computational efficiency comparison.** We compare parameter size (B), inference throughput (tokens per second) and training costs ($10^{18}$ FLOPs).

| Model | Size | Thrpt. | Train. Cost |
|---|---|---|---|
| LLaVA-NeXT-Qwen3-4B | 4.7 | 161.1 | 1.27 |
| DPA-Qwen3-4B | 5.5 | 151.8 | 1.45 |
| *ratio* | 1.17× | 0.94× | 1.14× |
| LLaVA-NeXT-Qwen3-32B | 33.4 | 57.8 | 9.33 |
| DPA-Qwen3-32B | 34.2 | 56.4 | 9.50 |
| *ratio* | 1.02× | 0.98× | 1.02× |

do not use Qwen3 4B or Qwen3 32B as the target large language model since MIR requires dimensions of both spaces to be the same. We observe that regardless of the training data scale, all VLMs effectively reduces the modality gap as the visual feature propagates through its layers, converging to a uniformly low level at the final output layer. Interestingly, perceivers trained with more data initially exhibit a larger gap in shallow layers, yet the deep language blocks successfully bridges these gaps. This confirms that VLMs as perceiver could reliably resolve the modality mismatch before the features ever reach the target large language model.

**Offloading Alignment Burden from the Target LLM.** The consequence of this pre-alignment is immediately visible in the target large language model internal space. Figure 5 compares the modality gap within the LLM layers from DPA models and controlled baselines. The result shows that DPA significantly reduce the modality gap, validating our core hypothesis that DPA effectively offloads the alignment workload and allows the target large language model to start understanding and reasoning on a more text native state. Furthermore, we find such modality gap advantage is consistent during the training process as shown in Figure 6.

**Geometric Isomorphism.** Beyond simple distance, we further explore the structural topology of the feature spaces in Figure 4 using heatmaps of intra-modal similarity, where each grid represents the distance between two internal layers. We find that the visual space structure of DPA models bears a resemblance to the text space structure, exhibiting shared "block-diagonal" subspaces. In contrast, the baseline's visual topology is divergent and fuzzy compared to text space and our visual space. This suggests that DPA achieves a form of geometric isomorphism: the visual features are not just "close" to text, but are organized with a similar structural logic, enabling the LLM to process images using its native, pre-trained circuits without destructive adaptation.

## 4.5. Computational Efficiency

A potential concern with replacing a ViT encoder with a perceiver VLM is the increase in computation. We analyze this trade off in terms of parameter size, training costs and inference throughput.

**High Return Compared with Introduced Parameter.** While perceivers introduces more parameters, this increase is negligible in the context of large full model sizes. For example, our Qwen3-0.6B based perceiver introduces only 2% more parameter to the 32B model. In contrast, the performance gains supports such slight overhead is highly efficient as the AI2D and MATH-500 increases by 6.0 and 13.8 points separately. Moreover, we validate that such return is non-trivial: simply add randomly initialized LLM (i.e., *w/o* perceiver LM pre-training in Table 4) or MLP with 5 times larger hidden-dimension whose parameter-size is same as Qwen3-0.6B (i.e., *w/* large MLP in Table 4) both results significantly worse performance. Suggests that the effectiveness of DPA roots from the combination of language priors and architectural depth design.

**Comparable Training Costs and Same Throughput Level.** Critically, the perceiver only incurs limited computational cost during pre-fill phase and results zero impact on generation speed. We computes the training FLOPs of DPA models and baselines on 4B and 32B scales, results in Table 5 shows that DPA only incurs 2% training costs for the 32B scale, which is promising compared with its performance improvement. Further more, we also analyze the inference throughput, results demonstrate that the small pre-fill overhead is almost fully amortized during generation and the throughput of DPA models is same level (e.g., 94% on the 4B scale and 98% on the 32B scale) as the baseline.

## 5. Related Works

### 5.1. Visual Encoding of Vision Language Models

Visual encoding serves as a vital component in vision language models. Early works (Alayrac et al., 2022; Li et al., 2023; Zhu et al., 2023; Ye et al., 2023), represented by

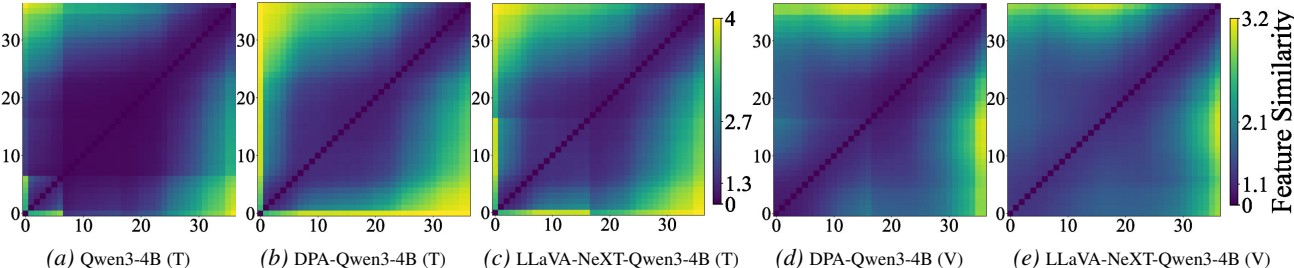

*Figure 4.* **Cross-layer intra-modal similarity matrices of text spaces and visual spaces.** "T" and "V" denote text and visual spaces, respectively. (Lighter colors indicate higher similarities.) The DPA visual space (d) exhibits "block-diagonal" subspaces that resemble the subspaces found in text spaces (a-c), whereas the baseline visual space (e) remains fuzzy.

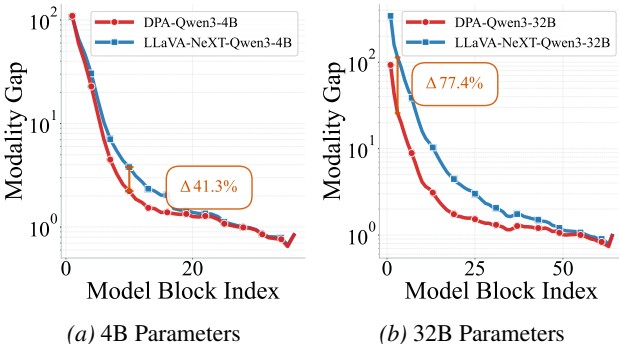

*Figure 5.* **Per-layer modality gap comparison of different models.** DPA consistently minimizes the modality gap on most layers, and the reduction on the 32B setting is more significant.

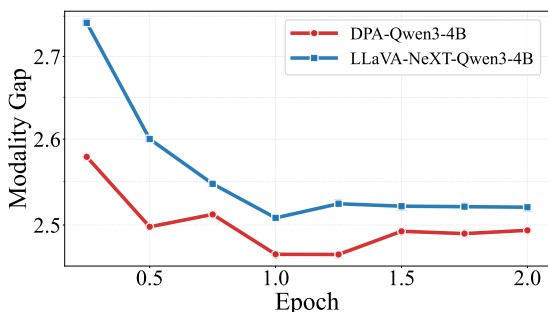

*Figure 6.* **Modality Gap dynamics during instruction tuning stage.** DPA consistently achieves smaller modality gap during the whole training process.

LLaVA (Liu et al., 2023; 2024a), use a fixed-resolution CLIP (Radford et al., 2021) directly as the visual encoder. The output visual features are then injected into the large language model through different connectors. Subsequent works (Liu et al., 2024b; Guo et al., 2024; Li et al., 2025) explore high-resolution visual understanding via encoding small image slices separately. Meanwhile, another thread of works combines multiple visual encoders such as DINO (Caron et al., 2021)and SAM (Kirillov et al., 2023) together with CLIP ViT encoders to grab a more comprehensive visual representation for input image (Tong et al.,

2024; Lin et al., 2023). Despite significant advances, visual features generated by existing methods still stay distant from the text space of text space of the target large language models (Huang et al., 2025; Shukor & Cord, 2024), wasting critical depth of target large language models on superficial modality alignment objective. On the contrary, our proposed DPA precisely targeting this problem by using a perceiver VLM to conduct deep pre-alignment for visual features before they entering the target large language models.

### 5.2. Text Capability Preservation

Catastrophic forgetting of text capabilities remains a primary bottleneck in developing robust vision language models. The prevailing solution is data-centric, focusing on optimizing the mixing ratios between text-only and multimodal corpora (Bai et al., 2025b; McKinzie et al., 2024). For instance, DeepSeek-VL (Lu et al., 2024a) gradually improve the multimodal data weight during training and keep the weight of text-only data above a threshold during multimodal training stages. Furthermore, InternVL (Chen et al., 2024b) explores mixing multimodal data into the text-only pre-training stage to balance the updates from different modalities from the beginning. However, these approaches essentially frame text capability preservation as a multi-task optimization trade-off, where the model must "repair" the forgetting caused by visual alignment via constant text replay. In contrast, our approach targets the structural root cause of the conflict rather than its symptoms. Crucially, as an architectural intervention, DPA is orthogonal to data engineering strategies and can be seamlessly combined with existing mixing recipes for maximum effect.

### 6. Conclusion

In this work, we introduce the Deep Pre-Alignment architecture for VLMs, which replace standard ViT encoders with a small perceiver VLM to offload multimodal alignment burdens from the target LLM. As visual features are deeply aligned to text space, less destructive adaptation are required for the target LLM, helping retain the strong skills

obtained from language pre-training. Experimental results show that DPA consistently improve performance over 11 popular multimodal and text benchmarks on different model scales from 4B to 32B. We further show that these gains hold when changing the target large language model from Qwen3 to a LLaMA 3.2 model, indicating that DPA gives universal improvement regardless of LLM families. In conclusion, DPA offers a seamless upgrade path for current VLM development with only a modular replacement for the visual encoder and marginal computational overhead.

## Impact Statement

This paper presents work whose goal is to advance the field of machine learning, specifically by improving the multimodal performance and text capability preservation of VLMs. By reducing the need for destructive adaptation and deeply align visual features before passing them to the target large language model, our approach contributes to more efficient AI development. While our work shares the general societal implications of VLMs—such as the risk of bias inherent in web-scale training data—we do not believe this specific architectural proposal introduces unique ethical consequences that require further highlighting.

## Acknowledgment

This work was funded by the Shanghai Qi Zhi Institute Innovation Program (SQZ202410), the high-quality development project of MIIT and a grant from the Guoqiang Institute, Tsinghua University. This work is also supported by Tsinghua University - Keystone Electrical (Zhejiang) Co.,Ltd Joint Research Center for Embodied Multimodal Artificial Intelligence (JCEMAI).

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

# A. Implementation Details

The specific hyperparameter settings are shown in Table 6. In stage-2, 32B models are trained using LoRA due to resource limitations and the epoch is changed to 3 for better convergence.

*Table 6.* **Hyperparameter settings.**

| Hyperparameter | Stage-1 | Stage-2 Full | Stage-2 LoRA |
|---|---|---|---|
| ***Model and Optimization Strategy*** | | | |
| Trainable Parameters | Projection | Full | LoRA weights |
| ***Learning Rate and Scheduler*** | | | |
| Learning Rate (ViT) | - | 2e-6 | 4e-5 |
| Learning Rate (LLM/Proj) | 1e-3 | 1e-5 | 2e-4 |
| LR Scheduler | Warmup Stable Decay | Cosine | Cosine |
| Warmup Steps | 440 | 500 | 500 |
| Weight Decay | 0.01 | 0.01 | 0.01 |
| ***Training Dynamics*** | | | |
| Num Epochs | 2 | 2 | 3 |
| Batch Size | 8 | 8 | 8 |
| Packing Length | 14,500 | 16,082 | 16,082 |

# B. Detail Experimental Results

We report detailed per-benchmark evaluation performance of our analysis experiments in this section.

*Table 7.* **Perceiver standalone evaluation (Table 2).** Best results highlighted in **bold**.

| Stage-1 | Stage-2 | MMVet | MMStar | Seed Bench2+ | Avg. | MMMU | Math Vista | Math Vision | Avg. | OCR Bench | AI2D | Avg. | MATH 500 | MMLU Redux | GPQA Diamond | Avg. | *Multi. Avg.* | *All Avg.* |
|---|---|---|---|---|---|---|---|---|---|---|---|---|---|---|---|---|---|---|
| × | × | 0.0 | 0.0 | 0.0 | 0.0 | 0.0 | 0.0 | 0.0 | 0.0 | 0.0 | 0.0 | 0.0 | **48.4** | 43.0 | **27.8** | 39.7 | 0.0 | 10.8 |
| ✓ | × | 4.4 | 15.9 | 7.0 | 9.1 | 11.9 | 21.8 | 9.0 | 14.2 | 19.7 | 19.8 | 19.8 | 51.6 | 43.8 | 26.8 | **40.7** | 13.7 | 21.1 |
| ✓ | 10k | 9.3 | 34.8 | 28.6 | 24.2 | 28.7 | 24.9 | 15.2 | 22.9 | 26.8 | 34.3 | 30.6 | 21.8 | **44.3** | 26.3 | 30.8 | 25.3 | 26.8 |
| ✓ | 250k | 17.2 | 34.4 | 40.6 | 30.7 | 30.2 | 30.1 | **16.3** | 25.5 | 46.2 | 44.4 | 45.3 | 11.2 | 37.7 | 23.7 | 24.2 | 32.4 | 30.2 |
| ✓ | 1M | **20.9** | **38.4** | **46.1** | **35.1** | 32.0 | **34.8** | 15.0 | **27.3** | 57.1 | 50.1 | 53.6 | 9.6 | 38.0 | 20.7 | 22.8 | **36.8** | **33.0** |

*Table 8.* **Performance of models using different perceivers (Table 3).** We report detailed results on our DPA-Qwen3-4B model. Best results highlighted in **bold**.

| Stage-1 | Stage-2 | MMVet | MMStar | Seed Bench2+ | Avg. | MMMU | Math Vista | Math Vision | Avg. | OCR Bench | AI2D | Avg. | MATH 500 | MMLU Redux | GPQA Diamond | Avg. | *Multi. Avg.* | *All Avg.* |
|---|---|---|---|---|---|---|---|---|---|---|---|---|---|---|---|---|---|---|
| × | × | 45.6 | 50.4 | 63.4 | 53.1 | **50.4** | 49.6 | 20.6 | 40.2 | 69.7 | 69.2 | 69.5 | 60.6 | **70.4** | 34.3 | 55.1 | 52.4 | 53.1 |
| ✓ | × | **47.8** | 50.9 | 62.4 | **53.7** | 49.8 | 52.5 | **21.9** | 41.4 | 70.8 | 69.6 | 70.2 | 54.8 | 70.1 | 35.4 | 53.4 | 53.2 | 53.3 |
| ✓ | 10K | 44.4 | 51.5 | 62.6 | 52.8 | 48.6 | **53.2** | 21.3 | 41.0 | 72.0 | 69.7 | 70.9 | **63.0** | 68.0 | 37.9 | **56.3** | 52.9 | 53.8 |
| ✓ | 250K | 45.5 | **51.6** | 63.3 | 53.5 | 50.3 | 52.9 | 21.8 | **41.7** | 71.6 | **70.4** | 71.0 | 60.2 | 69.3 | 37.9 | 55.8 | **53.4** | **54.1** |
| ✓ | 1M | 42.7 | 50.7 | **64.0** | 52.5 | 50.0 | 51.8 | 21.2 | 41.0 | **74.7** | 70.0 | **72.4** | 54.2 | 70.2 | 33.3 | 52.6 | 53.1 | 53.0 |

*Table 9.* **Ablation on perceiver components and design strategies (Table 4).** Best results highlighted in **bold**.

| Method | General | | | | Reasoning | | | | Perception | | | Text | | | | Multi. | All |
|---|---|---|---|---|---|---|---|---|---|---|---|---|---|---|---|---|---|
| | MMVet | MMStar | Seed Bench2+ | Avg. | MMMU | Math Vista | Math Vision | Avg. | OCR Bench | AI2D | Avg. | MATH 500 | MMLU Redux | GPQA Diamond | Avg. | Avg. | Avg. |
| LLaVA-NeXT-Qwen3-4B | 41.4 | 50.7 | 61.2 | 51.1 | 49.9 | 49.6 | 20.7 | 40.1 | 69.1 | 67.4 | 68.3 | 36.4 | 66.0 | 32.8 | 45.1 | 51.2 | 49.6 |
| w/ large MLP | 8.9 | 31.1 | 38.5 | 26.2 | 39.8 | 30.3 | 19.0 | 29.7 | 1.6 | 56.7 | 29.2 | 43.6 | 68.0 | **37.9** | 49.8 | 28.2 | 34.1 |
| DPA-Qwen3-4B | 42.7 | 50.7 | **64.0** | 52.5 | 50.0 | 51.8 | 21.2 | 41.0 | **74.7** | 70.0 | **72.4** | 54.2 | 70.2 | 33.3 | 52.6 | 53.1 | 53.0 |
| w/o perceiver LM blocks | 42.1 | 51.3 | 61.7 | 51.7 | 47.1 | 52.6 | 21.1 | 40.3 | 70.9 | 68.1 | 69.5 | 35.2 | 65.3 | **37.9** | 46.1 | 51.9 | 50.3 |
| w/o perceiver LM pre-training | 14.5 | 34.0 | 42.6 | 30.4 | 42.0 | 35.4 | 20.5 | 32.6 | 3.0 | 61.3 | 32.2 | 62.2 | **72.4** | **37.9** | 57.5 | 31.7 | 38.7 |
| w/ instruction context | 47.7 | 51.1 | 62.5 | 53.8 | 50.1 | 52.6 | **22.6** | **41.8** | 72.5 | **71.0** | 71.8 | **67.8** | 71.9 | 37.3 | **59.0** | 53.8 | **55.2** |
| w/ perceiver frozen | 43.1 | 51.3 | 60.7 | 51.7 | 48.6 | 49.9 | 21.3 | 39.9 | 67.0 | 67.8 | 67.4 | 61.6 | 71.9 | 29.8 | 54.4 | 51.2 | 52.1 |
| w/ untrained perceiver | 45.6 | 50.4 | 63.4 | 53.1 | **50.4** | 49.6 | 20.6 | 40.2 | 69.7 | 69.2 | 69.5 | 60.6 | 70.4 | 34.3 | 55.1 | 52.4 | 53.1 |
| w/ untrained perceiver + multitask | **49.6** | **52.6** | 63.6 | **55.3** | 47.9 | **53.9** | 22.2 | 41.3 | 72.1 | 69.9 | 71.0 | 57.6 | 71.3 | 31.8 | 53.6 | **54.0** | 53.9 |

## C. Quantifying Destructive Adaptation

*Table 10.* **Analysis of destructive adaptation.** We compare the Update Density (proportion of parameters with changes large than 1e-6) and Intrusion Dimension (number of singular vectors with minimum cosine similarity to original weights smaller than 0.9) in the target LLM. DPA consistently yields lower values, indicating sparser and less intrusive updates.

| Model | Update Density ↓ | Intrusion Dimension ↓ |
|---|---|---|
| LLaVA-NeXT-Qwen3-4B | 0.932 | 1970 |
| **DPA-Qwen3-4B** | **0.926** | **1693** |
| LLaVA-NeXT-Qwen3-32B | 0.958 | 30705 |
| **DPA-Qwen3-32B** | **0.931** | **29223** |

We analyze the weight updates after stage-2 to quantify the destructive adaptation. We employ two metrics: (1) **update density** measures the proportion of parameters in the target LLM that undergo significant updates. A lower density indicates a sparser updates and the that the model retains more of its pre-trained state; (2) **intrusion dimension** (Shuttleworth et al., 2025) measures the number of singular vectors in trained weights that are not similar to any singular vector from untrained weights. As shown in Table 10, we observe that DPA consistently exhibits sparser updates and lower intrusion dimension compared to the baseline on both the 4B and the 32B scale. These results validate that DPA effectively mitigates destructive adapation for the target LLM.

## D. Failure Behaviors and Causes of Cambrian-1 on Text-Only Reasoning

In Table 1, Cambrian-1-8B (Tong et al., 2024) shows near-zero scores on text-only reasoning benchmarks (MATH-500 / MMLU-Redux / GPQA-Diamond), despite being initialized from LLaMA-3-instruct-8B(Meta, 2024). We find the degradation is dominated by generation collapse, suggesting catastrophic forgetting of instruction-following and long-form reasoning.

### D.1. Error distribution of text-only evaluations

We categorize generations into: **empty** (blank or only whitespace), **degenerate** (nonsensical or invalid format), and **parsable** (valid format with content). Table 11 showcases the real counts/ratios.

*Table 11.* **Cambrian-1 output distribution on text benchmarks.**

| Benchmark | Empty | Degenerate | Parsable |
|---|---|---|---|
| MMLU-Redux (2779) | 2090 (75.21%) | 112 (4.03%) | 577 (20.76%) |
| GPQA-Diamond (198) | 197 (99.49%) | 0 (0.00%) | 1 (0.51%) |
| MATH-500 (500) | 391 (78.20%) | 52 (10.40%) | 57 (11.40%) |

### D.2. Likely causes

We hypothesize two key factors:

**Language-only datasets included in multimodal training are weak and lacks long/hard reasoning.** During instruction tuning, Cambrian-1 uses only a limited portion of language-only data (e.g., Dolly, MathInstruct, WizardCoder, OrcaMath, OpenCodeInterpreter, OpenOrca), and does not include hard math reasoning datasets or long-CoT trajectories, making long-form text reasoning effectively OOD.

**Instruction-following collapse at decoding time.** The extreme empty-output rates (e.g., 99.49% on GPQA) suggest the model frequently terminates early or fails to produce the required answer format, which is consistent with catastrophic forgetting of core LLM behaviors under multimodal fine-tuning.

### D.3. Case study

---

**Prompt**

```
Answer the following multiple choice question. The last line of your response should
↪   be of the following format: 'ANSWER: $LETTER' (without quotes) where LETTER is
↪   one of ABCD. Think step by step before answering.

Predict the hyperfine value for the EPR spectrum of fully deuteriated benzene
↪   radical anion C6D6*-.

A) 0.375 mT
B) 3.75 G
C) 2.35 mT
D) 0.58 G
```

---

**Prediction**

```
[BOS] Js) = 1.4 × 10^-4 T

4. Calculate the hyperfine value: The hyperfine value is the product of the
↪   hyperfine coupling constant and the number of hyperfine lines. In this case, the
↪   hyperfine value is:

   1.4 × 10^-4 T × 12 = 1.68 × 10^-4 T

5. Convert the hyperfine value to the given format: The hyperfine value is 1.68 ×
↪   10^-4 T, which is equivalent to 1.68 × 10^-4 mT. To convert this to the given
↪   format, we can multiply by 10^3:

   1.68 × 10^-4 mT × 10^3 = 1.68 × 10^-1 mT

6. Match the hyperfine value to the given options: The hyperfine value is 1.68 ×
↪   10^-1 mT, which is closest to 0.375 mT.

Therefore, the answer is 0.375 mT.
```

---

The logical reasoning in the rationale makes no sense at all and does not follow the required `ANSWER: X` format. The beginning of the prediction is also incomplete.

