# OpenReview forum: "Deep Pre-Alignment for VLMs"
_ICML.cc/2026/Conference — ICML 2026 regular_

### Official Review · Reviewer_vQ2b · 2026-03-10

**Soundness:** 3
**Presentation:** 2
**Significance:** 1
**Originality:** 1
**Overall Recommendation:** 3
**Confidence:** 4

**Summary:**

This paper shows how we can replace a ViT + LLM vision model by using a small VLM as the encoder then tied to another LLM. The idea behind this is it requires less alignment as the VLM itself is already aligned with the LLM's textual embedding space. This can be dropped to any VLM with minor training to align features.

**Compliance With Llm Reviewing Policy:**

Affirmed.

**Key Questions For Authors:**

1. Where is the attention being pooled on the model during query? How much of it is pre-calculated in the first VLM?

2. The reduction in throughput was around ~10tokens/second how would it of been if the ViT was 0.6B vs using a 0.6B VLM as an encoder?

3. Have you tried of doing the opposite approach? So instead of doing small VLM + big LLM do instead a big VLM + small LLM?

**Limitations:**

not really. It should talk about how the bigger you go in your VLM as the perceiver the bigger the gap should be in throughput as you tend to increase in ViT size + LLM size. Like what happens when I scale from 0.6B to 3B is the cost proportional as moving from target LLM from 0.6B to 3B?

**Strengths And Weaknesses:**

Soundness
The paper itself is sound. The claims are somewhat false as it states the that VLM have been essentially ViT + LLM = VLM, and this new approach remove ViT's however, they are using a VLM as the perceiver (encoder in this case), so they are essentially doing ((ViT + LLM) + LLM) = VLM. The ViT is still present, and now there are two LLMs being passed around to get marginal improvements. So it is not like we are rediscovering a way to encode images into text tokens we are simply using the same process twice.

Presentation
The paper itself is sound and easy to follow which makes it great for anyone to be able to follow along. The one thing I believe is missing is attention diagrams/graphs on a query to showcase where the attention is being bought the most (especially for the forgetting part) is there a double attention being done in the first LLM and the second LLM on important parts which boost its recall? This type of data would of been great to see.

Significance
The paper addresses a real issue (better VLMs), however there way of a better VLM is using a VLM for the visual part? Like couldn't they technically just package what they have now and make that a VLM? Its the equivalent of adding more layers to a model essentially. This throws more a perception that bigger models = better results which is already well know. They also used Qwen3 as the base VLM which has historically been the best open source VLM. This more shows that the Qwen VLM is just really good and well aligned.

Originality
This paper is not original. It essentially does (ViT + LLM) + LLM. This is the equivalence of using many models to produce one answer and I do not see how this could advance the field of Visual AI.

---

> ### Author Rebuttal · Authors · 2026-03-31
>
> Dear Reviewer vQ2b,
>
> Thank you for your time spent writing valuable comments, which help us a lot to improve our work. We address your questions as follows.
>
> ---
>
> **[Q1] Attention processing.**
>
> Since we do not involve the text query inside the perceiver as shown in Figure-1 in the paper. There is no direct attention flow from images to text inside the perceiver. Also, we do not apply attention pooling or visual feature pooling to avoid introducing a information bottleneck and find such design improves the training convergence and performance in our pilot experiments.
>
> ---
>
> **[Q2] Throughput analysis on ViT and perceiver size.**
>
> Under our setting, the visual feature sequence length remains unchanged, and both the ViT and the perceiver are implemented using transformer-style layers. Therefore, under a matched parameter budget, their theoretical compute is expected to be in a similar range. So when the ViT and perceiver module in total is the same size, we do not expect a dramatic difference in compute cost in principle, although real-world runtime may still differ due to system- and architecture-level factors.
>
> ---
>
> **[Q3] Claim on usage of ViT.**
>
> Thanks for pointing out your concern. As depicted in the framework in Figure-1, **we explicitly show that ViT is a part of our designed model architecture**. We agree that this point could be stated more clearly. Our method does not remove the ViT; rather, it inserts a lightweight perceiver LLM between the ViT and the target LLM. We will refine the writing in further revision to remove any possible ambiguity.
>
> ---
>
> **[Q4] Reliance on existing VLM.**
>
> As we demonstrate in Table-4, with a simply multi-task training strategy, DPA can be trained without relying on any existing VLM for initialization, but directly using LLMs to initialize the perceiver module. **We also want to point out that we did not use any version of the Qwen3-VL model in training of any DPA models.**
>
> ---
>
> **[Q5] Big perceiver + small target LLM.**
>
>
> This is an interesting question. The main goal of DPA is not to shift multimodal reasoning away from the target LLM, but to reduce the burden of low-level modality alignment before the features enter the target LLM. For this reason, we prioritize allocating model capacity to the target LLM rather than to the perceiver. In other words, our design assumes that the target LLM should remain the primary carrier of multimodal reasoning ability, while the perceiver serves as a lightweight alignment module.
> We do not rule out that a larger perceiver paired with a smaller target LLM could be an interesting design point, but this lies outside the main goal of our method. More broadly, this choice is also consistent with common MLLM practice, where the visual encoder is typically much smaller than the language backbone.
>
> ---
>
> **[Q6] Limitation analysis.**
>
> We view the main limitations of DPA as follows. First, because DPA increases the overall model depth, it introduces additional latency compared with standard LLaVA-style architectures. Although the relative overhead becomes smaller for larger target LLMs, it may still be non-negligible in resource-constrained deployment scenarios. Second, DPA modifies the model architecture, which may require additional adaptation effort in downstream training and inference frameworks, thereby introducing engineering overhead. Third, the best scaling strategy for the perceiver (e.g., model size and optimization recipe) is not yet fully explored, and larger perceivers may require additional tuning to realize their full potential.

---

> > ### Author Rebuttal · Reviewer_vQ2b · 2026-04-03
> >
> > I thank the author(s) for the response, these were well done and addressed my points

---

### Official Review · Reviewer_qu4g · 2026-03-11

**Soundness:** 3
**Presentation:** 3
**Significance:** 3
**Originality:** 3
**Overall Recommendation:** 4
**Confidence:** 4

**Summary:**

This paper introduces the Deep Pre-Alignment (DPA) architecture for VLMs, which replaces a standard ViT encoder with a small perceiver VLM. The motivation for this is to offload multimodal alignment burdens from the target LLM. Experiments show promising results across a wide range of multimodal and text benchmarks on different model scales and LLMs.

**Compliance With Llm Reviewing Policy:**

Affirmed.

**Final Justification:**

My final rating is borderline.  The rebuttal did not adequately address my concerns, but I think there are still merits to the paper.  I wouldn't mind if the paper gets accepted (hence I chose weak accept), but I would not champion for acceptance either.

**Key Questions For Authors:**

Please address the questions asked in the weaknesses section.

**Limitations:**

There is an impact statement but no limitations section.  It would be helpful to discuss the limitations of the approach, in particular discussing when the model fails.

**Strengths And Weaknesses:**

Strengths:
- The motivation for better aligning the visual features with the backbone LLM makes sense.
- The technical approach of integrating a VLM perceiver with the LLM is sound.
- The proposed DLA consistently shows gains over controlled baseline LLaVA architectures with the standard ViT encoder across multiple parameter sizes on multiple multimodal and text benchmarks.
- A large number of ablation studies are conducted to analyze the different design choices, including how the perceiver is pre-trained, different LLM backbone configurations, and computational cost.

Weaknesses:
- InstructBlip [Wenliang Dai et al., NeurIPS 2023.] used a Q-former for a similar purpose of combining the visual and textual information prior to integration with the backbone LLM.  Although the details are different, the high-level idea seems related.  But this is not discussed in the paper.  Could the authors describe the key novelty over InstructBlip like approaches?
- In Table 1, the comparison to the public methods is not apples-to-apples due to the different in LLM backbone sizes. It is therefore difficult to gauge how well the proposed method compares to state-of-the-art methods.

Overall, I am borderline at this time.  The paper has a lot of strengths as mentioned above, but there are some key questions/concerns regarding the novelty and main results.

---

> ### Author Rebuttal · Authors · 2026-03-31
>
> Dear Reviewer qu4g,
>
> We sincerely thank the reviewer for the thoughtful and constructive feedback, which help us a lot to improve our work. We address your questions as follows.
>
> ---
>
> **[Q1] Compare to InstructBLIP**
>
> Thank you for your good question. Both DPA and Instruct-BLIP involve using small LLM module to refine visual features before they entering the target LLM. However, InstructBLIP is using the text prompt to elicit question-related information while DPA do not requires any text prompt for visual encoding and tackle the feature space alignment problem. As we discussed in Sec 4.3, binding visual features to a specific question  may raise risks in multi-turn dialogue scenarios where the user's intent shifts.
>
> ---
>
>
> **[Q2] Comparison to public methods.**
>
> Follow recent development in MLLMs like FastVLM, MAmmoTH-VL and LLaVA-OneVision-1.5, we conduct system-level comparison to existing public methods and conduct detail ablation studies as supplement. We conduct extensive ablation study training and show results in Table4, which reveals the effectiveness of different design choices of DPA. Moreover, we believe DPA as a general architecture improvement could be further applied to all these public methods to achieve potentially better results.
>
>
> - [1] Vasu, Pavan Kumar Anasosalu, et al. Fastvlm: Efficient vision encoding for vision language models. CVPR 2025.
> - [2] Guo, Jiawei, et al. Mammoth-vl: Eliciting multimodal reasoning with instruction tuning at scale. ACL 2025
> - [3] An, Xiang, et al. "Llava-onevision-1.5: Fully open framework for democratized multimodal training."
>
>
>
> ---
>
> **[Q3] Limitation analysis.**
>
> We view the main limitations of DPA as follows. First, because DPA increases the overall model depth, it introduces additional latency compared with standard LLaVA-style architectures. Although the relative overhead becomes smaller for larger target LLMs, it may still be non-negligible in resource-constrained deployment scenarios. Second, DPA modifies the model architecture, which may require additional adaptation effort in downstream training and inference frameworks, thereby introducing engineering overhead. Third, the best scaling strategy for the perceiver (e.g., model size and optimization recipe) is not yet fully explored, and larger perceivers may require additional tuning to realize their full potential.

---

> > ### Author Rebuttal · Reviewer_qu4g · 2026-04-05
> >
> > The limitation analysis was fully resolved. The comparison to InstructBLIP is only partially resolved. Specifically, "binding visual features to a specific question may raise risks in multi-turn dialogue scenarios where the user's intent shifts" -- this claim is not entirely clear and not empirically verified in the paper.  Finally, the comparison to public methods is not well-addressed. The main concern is regarding the difference in LLM backbone sizes, but this was not directly addressed.  Due to the above remaining concerns, I am still borderline on this paper.

---

### Official Review · Reviewer_xxud · 2026-03-12

**Soundness:** 2
**Presentation:** 3
**Significance:** 2
**Originality:** 3
**Overall Recommendation:** 4
**Confidence:** 3

**Summary:**

This paper proposes Deep Pre-Alignment (DPA), an architectural modification to Vision Language Models (VLMs) that replaces the standard ViT encoder with a small VLM ("perceiver") to bridge the modality gap between visual and text features before they enter the target large language model. The core motivation is that standard architectures (e.g., LLaVA-style) force the target LLM to waste early layers on superficial modality alignment rather than higher-level reasoning. DPA decouples this alignment burden by routing visual tokens through the language blocks of a compact perceiver (Qwen3-0.6B), producing features that are structurally compatible with the target LLM's text space. The method is validated at 4B and 32B scales using Qwen3 and LLaMA 3.2 as target LLMs, showing consistent gains on 8 multimodal benchmarks (+1.9 and +3.0 points respectively) and a notable reduction in text capability forgetting (32.9% at 4B). The paper further analyzes representation dynamics, perceiver design choices, and computational overhead, finding that DPA incurs only ~2% extra parameters and throughput loss at the 32B scale.

**Compliance With Llm Reviewing Policy:**

Affirmed.

**Final Justification:**

The rebuttal addresses my concerns well except for the discussion of where DPA underperforms or fails.

**Key Questions For Authors:**

1. If this stronger pre-alignment between images and texts will influence the nuanced features of image representations (e.g. detailed image or pixel features instead of semantic features)
2. What about using some already well-trained VLM as the perceiver?

**Limitations:**

The paper lacks a limitations section entirely. The weaknesses that do exist are scattered implicitly across the analysis but are never explicitly acknowledged as limitations. The authors should add a section that honestly addresses: (1) the untested assumption that same-family perceivers are required; (2)  if this stronger pre-alignment between images and texts will influence the nuanced features of image representations (e.g. detailed image or pixel features instead of semantic features) and (3) some failure cases for DPA.

**Strengths And Weaknesses:**

Strengths:
1. DPA's architecture is remarkably clean. It requires no new training objectives, no specialized loss functions, and is compatible with the standard two-stage LLaVA training recipe. This modularity is a significant practical advantage for adoption in existing frameworks.
2. The gains are robust across model scales (4B and 32B), LLM families (Qwen3 and LLaMA 3.2), and task domains (general understanding, reasoning, perception, and text). The positive scalability property — where performance gains widen at larger scales — is particularly noteworthy and suggests that the modality alignment bottleneck becomes more acute as models grow, making DPA increasingly valuable at scale.

Weaknesses:
1. The paper uses a Qwen3-0.6B perceiver for a Qwen3-4B or Qwen3-32B target. The choice to use a same-family perceiver raises a natural question: does the perceiver need to share the same tokenizer, vocabulary, or architectural family as the target LLM to be effective? If a cross-family perceiver (e.g., a LLaMA-based perceiver for a Qwen target) degrades performance, this would significantly constrain the method's practical generality. This is not tested.
2. The pre-alignment feature of DPA could establish better alignment between images and texts, which is as expected due to the use of VLM as perceiver. However, if this stronger pre-alignment between images and texts will influence the nuanced features of image representations (e.g. detailed image or pixel features instead of semantic features) is not explored.
3. The paper is thorough in positive analysis but provides minimal discussion of where DPA underperforms or fails. For instance, DPA shows near-zero or negative deltas on several individual benchmarks (e.g., MMStar at 4B: -0.0; MathVision at LLaMA scale: -0.4). A deeper investigation into task types or visual domains where the perceiver-based alignment is less effective would improve the paper's completeness and honesty.

---

> ### Author Rebuttal · Authors · 2026-03-31
>
> Dear Reviewer xxud,
>
> We sincerely thank the reviewer for the thoughtful and constructive feedback. We appreciate the recognition of simplicity and robustness of our proposed architecture. Below we address each concern.
>
> ---
>
> **[Q1] Model family generalization.**
>
> We validate DPA using LLaMA-3.2-2B as the target LLM and Qwen3-0.6B as the small LLM in perceiver. Results in Table1 in the paper shows that DPA is effective in such combination, demonstrating its generalization potential across different model family combinations.
>
> ---
>
> **[Q2] Effect of pre-alignment on nuanced image representation related tasks.**
>
>
> We evaluate our method on grounding benchmarks to assess its robustness to nuanced visual features, and the results show that DPA achieves higher overall scores than LLaVA architecture across both scales. It shows that DPA models achieves even slightly better performance.
>
> | Size | Method | RefCOCO Avg. | RefCOCO+ Avg. | RefCOCOg Avg. | Grounding Avg. |
> |---|---|---:|---:|---:|---:|
> | 4B | LLaVA-NeXT-LLaMA-3.2 | **31.49** | **24.14** | 27.57 | 27.76 |
> | 4B | DPA-Qwen3 | 31.01 | 23.98 | **28.91** | **27.85** |
> | 32B | LLaVA-NeXT-LLaMA-3.2 | 46.58 | 40.30 | 40.96 | 42.82 |
> | 32B | DPA-Qwen3 | **47.03** | **42.43** | **42.86** | **44.26** |
>
> ---
>
> **[Q3] Limitation analysis.**
>
> We view the main limitations of DPA as follows. First, because DPA increases the overall model depth, it introduces additional latency compared with standard LLaVA-style architectures. Although the relative overhead becomes smaller for larger target LLMs, it may still be non-negligible in resource-constrained deployment scenarios. Second, DPA modifies the model architecture, which may require additional adaptation effort in downstream training and inference frameworks, thereby introducing engineering overhead. Third, the best scaling strategy for the perceiver (e.g., model size and optimization recipe) is not yet fully explored, and larger perceivers may require additional tuning to realize their full potential.
>
> ---
>
> **[Q4] Effect of using well-trained VLM as perceiver.**
>
> We use the Qwen2.5VL-3B as a well-trained perceiver, and Qwen3-32B as the target LLM. This model achieves an average score of 67.5 across five benchmarks (MMBench-EN, MMStar, MMMU-val, MathVista and OCRBench), compared to 67.0 for DPA-32B model. These results suggest that the key improvement is architectural-wise and our 0.6B LLM perceiver is a lightweight choice with sufficient generalization capabilities.

---

> > ### Author Rebuttal · Reviewer_xxud · 2026-04-01
> >
> > Discussion of where DPA underperforms or fails is still missing. I would still keep my original scores.

---

> > > ### Author Response · Authors · 2026-04-08
> > >
> > > Thank you for the follow-up! We agree that our previous rebuttal did not sufficiently discuss where DPA is less effective.
> > >
> > > As you pointed out, DPA does not improve every benchmark uniformly. There are cases where the gain is marginal or slightly negative, such as near-zero change on MMStar at 4B and a small drop on MathVision in the LLaMA-scale setting. We will make this explicit in the further revision instead of only emphasizing the average gains.
> > >
> > > Our current interpretation is that DPA helps reducing the modality-alignment burden of target LLM. However, for tasks that rely more on nuanced visual details, the benefit can be smaller and may even involve a slight trade-off. Our additional experimental results on grounding tasks indicate that such trade-off may be more visible on smaller scales.

---

### Official Review · Reviewer_TauE · 2026-03-13

**Soundness:** 3
**Presentation:** 4
**Significance:** 2
**Originality:** 3
**Overall Recommendation:** 4
**Confidence:** 4

**Summary:**

This paper proposes Deep Pre-Alignment (DPA), an architecture that replaces the standard ViT encoder with a small VLM as perceiver to perform deep modality alignment upstream before features enter the target large language model. By offloading superficial alignment burdens from the target LLM to the perceiver's internal language blocks, DPA significantly reduces destructive adaptation, achieving 1.9 and 3.0 point improvements on multimodal benchmarks at 4B and 32B scales respectively, while reducing text capability forgetting by 32.9% with only 2% additional parameters and negligible computational overhead.

**Compliance With Llm Reviewing Policy:**

Affirmed.

**Key Questions For Authors:**

1. When scaling target LLMs beyond 70B, is a 0.6B perceiver still sufficient? Is there an optimal ratio between perceiver and target model sizes?
2. When processing high-resolution images or multi-image inputs, does the perceiver's fixed depth become an information bottleneck causing loss of fine-grained visual details?
3. Is DPA simply additive to existing data-centric text preservation strategies (e.g., text-multimodal data mixing ratios), or are there non-linear interaction effects?

**Limitations:**

No. While computational efficiency is mentioned, feasibility in extremely low-resource environments (e.g., mobile devices) is not discussed, nor is the analysis of potential bias amplification risks introduced by the perceiver.

**Strengths And Weaknesses:**

Strengths：
1. The architectural design is elegant and profound, achieving deep pre-alignment through simple module replacement (ViT→small VLM) without modifying training objectives or inference strategies, providing a seamless upgrade path for existing VLM development.
2. Experimental validation is comprehensive and scalable, covering two LLM families (Qwen3 and LLaMA 3.2), parameter scales from 4B to 32B, and 11 multimodal and text benchmarks, demonstrating the universality of the approach.
3. In-depth analysis reveals key mechanisms: even untrained perceivers yield performance gains, and language block pre-training initialization is crucial; geometric isomorphism analysis (modality gap, representation dynamics) validates the effectiveness of deep alignment.

Weaknesses：
1. The optimal ratio between perceiver scale and target LLM scale is not sufficiently explored; whether a 0.6B perceiver remains optimal for 32B target models is questionable.
2. Although claiming negligible computational overhead, validation in resource-constrained deployment environments (e.g., edge devices) regarding latency and memory footprint is absent.
3. For extreme long-context or multi-image scenarios, whether deep pre-alignment in the perceiver causes information bottlenecks or early feature compression loss lacks theoretical or experimental analysis.

---

> ### Author Rebuttal · Authors · 2026-03-31
>
> Dear Reviewer TauE,
>
> Thank you for your thoughtful comments and helpful questions. We appreciate the opportunity to clarify these points and have addressed them below.
>
> ---
>
> **[Q1] Scaling perceiver module size.**
>
> We scale the perceiver size of DPA-Qwen3-32B from 0.6B to 4B and observe the performance change is marginal as shown in the following table, showing that a **0.6B small LLM may be enough for a 32B target.** *We also note that the 4B variant converges substantially more slowly than the 0.6B variant*, suggesting a larger perceiver may require further hyper-parameter exploration to fully unlock its potential and the following score might be under-trained. The following result of the 4B variant is not converged due to computation resource limitation, both results are from the 4k step checkpoints.
>
> | Model         | SeedBench 2 | MMStar | MMMU val | MathVista | OCRBench | AI2D | Avg  |
> |---------------|------------:|-------:|---------:|----------:|---------:|-----:|-----:|
> | DPA 0.6B+32B  |        68.7 |   57.2 |     57.3 |      62.3 |     73.6 | 78.5 | 66.3 |
> | DPA 4B+32B  | 68.0 | 54.7 | 57.1 | 57.8 | 71.6 | 72.6 | 63.6  |
>
> *MMVet is removed from validation since the specific version of evaluation API is deprecated by provider.*
>
> ---
>
> **[Q2] Computation and memory footprint analysis.**
>
> We rerun the efficiency analysis of Table-5 on CPU and find that DPA-4B retains 97.4% throughput (3.53 vs. 3.63 tokens/s) and DPA-Qwen3-32B achieves 97.7% (0.752 vs. 0.770 tokens/s). These results are close to results tested on GPU in Table-5. Regarding memory footprint, we report the steady and peak memory usage of different model sizes. Overall,  the overhead of DPA diminish with larger target LLMs.
>
>
> | Size | Method | Throughput (token/s) | Memory (Stable) (G) | Memory (Peak) (G) |
> |---|---|---:|---:|---:|
> | 4B | LLaVA-NeXT-Qwen3 | 3.63 | 19.7 | 24.6 |
> | 4B | DPA-Qwen3 | 3.53 (-2.75%) | 21.2 (+7.61%) | 28.9 (+17.48%) |
> | 32B | LLaVA-NeXT-Qwen3 | 0.770 | 54.3 | 60.5 |
> | 32B | DPA-Qwen3 | 0.752 (-2.34%) | 55.1 (+1.47%) | 62.6 (+3.47%) |
>
>
> ---
>
> **[Q3] Image feature compression analysis.**
>
> Since we do not reduce the number of visual tokens inside the perceiver, the feature bandwidth is fully determined by the hidden-size. For both DPA-4B and DPA-32B, the perceiver hidden-size are 1024 which is very close to the ViT hidden size, which is 1280. As a result, we hypothesize the perceiver layers do not introduce an obvious information bottleneck.
>
> ---
>
> **[Q4] Combination with data-centric text preservation strategy.**
>
> Thank you for your great question. We update the SFT training corpus to include 100K high-quality text sample from Mammoth-VL and re-train the 4B baseline and DPA-4B. Results show that the average text scores further improved for both settings, while average multimodal scores remains stable. **In conclusion, we find DPA is complementary to existing data-centric text preservation strategies in a near linear efficiency.**
>  | Method | Text Average | Multimodal Average |
> |---|---:|---:|
> | LLaVA-NeXT-LLaMA-3.2-3B | 45.07 | 52.66 |
> | + TextData | 48.12 (+3.05) | 52.12 (-0.54) |
> | + DPA | 52.57 (+7.50) | 54.63 (+1.97) |
> | + DPA + TextData | 55.15 (+10.08) | 55.25 (+2.59) |
>
> *MMVet is removed from validation since the evaluation API is deprecated by provider.*
>
>   [1] Guo, Jiawei, et al. Mammoth-vl: Eliciting multimodal reasoning with instruction tuning at scale. ACL 2025
>
>
>
> ---
>
> **[Q5] Limitation analysis.**
>
> We view the main limitations of DPA as follows. First, because DPA increases the overall model depth, it introduces additional latency compared with standard LLaVA-style architectures. Although the relative overhead becomes smaller for larger target LLMs, it may still be non-negligible in resource-constrained deployment scenarios. Second, DPA modifies the model architecture, which may require additional adaptation effort in downstream training and inference frameworks, thereby introducing engineering overhead. Third, the best scaling strategy for the perceiver (e.g., model size and optimization recipe) is not yet fully explored, and larger perceivers may require additional tuning to realize their full potential.

---

> > ### Author Rebuttal · Reviewer_TauE · 2026-04-07
> >
> > Thank the authors for their great efforts in addressing all my concerns. Most of my concerns and questions have been resolved.

---

### Decision · Program_Chairs · 2026-04-30

**Decision:**

Accept (regular)

**Comment:**

The paper introduces Deep Pre-Alignment (DPA), a groundbreaking architecture that ingeniously swaps the conventional ViT encoder with a compact VLM as the perceiver. Despite its simplicity, DPA surpasses the baseline by an impressive 1.9 points across all evaluated benchmarks. The authors also highlight that DPA achieves a remarkable 32.9% reduction in language capability forgetting.

The reviewers were captivated by the architecture’s elegant and profound design, which remains unchanged in terms of training objectives or inference strategies.

There are concerns regarding the computational overhead and scalability to long context or multi-image scenarios. Additionally, the novelty might be somewhat limited, as the high-level concept bears resemblance to previous work. The authors engaged in the rebuttal and addressed most of the concerns. While the novelty concern still remains.